# Gas network value chain optimization considering compressor capacity constraints

**Jinfeng Qiu**[1], **Liang Zhao**[2], **Xifeng Ning**[2], **Dejun Yu**[2]*

1 Algorithm Centre, UHAlean Information Technologies Co., Ltd., Shanghai, China, 2 Information Engineering, PetroChina Planning & Engineering Institute, Beijing, China

* yudejun@petrochina.com.cn

## Abstract

Efficient operation of natural gas pipeline networks is essential for minimizing costs and ensuring a stable energy supply. Compressor stations are critical for maintaining pressure throughout the network and represent a substantial portion of both capital investment and operational expenditures. Optimizing compressor operations is complex due to nonlinear interactions among compression ratios, flow rates, and operational constraints such as surge and choke limits, as well as the need for discrete decisions, including bypass selection and flow direction. To address these challenges, this study introduces a Sequential Linear Programming (SLP) algorithm specifically designed for large-scale pipeline scheduling. The method accommodates compressor capacity constraints and discrete decision variables, resulting in significant improvements in computational efficiency and solution quality. Numerical experiments on extensive real-world pipeline networks in China demonstrate that the algorithm rapidly produces near-optimal solutions, indicating its suitability for practical implementation. The results underscore clear advantages in both economic performance and operational reliability.

## 1. Introduction

Natural gas pipeline networks play a critical role in energy distribution, connecting production sources to a wide range of demand points—including industrial, commercial, and residential consumers. Optimizing the gas network value chain—which encompasses procurement, transportation, and distribution—is essential for achieving operational efficiency, cost-effectiveness, and supply reliability. Among these stages, pipeline transportation presents significant challenges due to inherent complexities, including pressure loss caused by frictional forces, fluctuating flow demands, and the physical properties of natural gas.

To maintain adequate gas pressure for efficient long-distance transport, compressor stations are strategically installed along pipelines. Often referred to as the "heart" of the pipeline network, compressor stations account for 20–30% of the total pipeline

**Data availability statement:** All data of gas networks are available from the github https://github.com/masonYau/gasNetworkData/tree/main.

**Funding:** The author(s) received no specific funding for this work.

**Competing interests:** The authors have declared that no competing interests exist.

infrastructure investment, with compressor units themselves comprising about half of that amount. In addition to their capital cost, compressor operations account for over 50% of total operational expenses, making them a critical factor in the economic performance of gas networks[25]. Therefore, accurately modeling and optimizing compressor operations—including their capacity limits—is not only beneficial for reducing costs but also essential for ensuring safety and operational efficiency.

Compressor capacity, typically expressed as a compression ratio, is highly sensitive to operating conditions, including inlet pressure, gas flow rate, and rotational speed. These relationships are inherently nonlinear and governed by complex thermodynamic and mechanical principles. Moreover, operational limits such as surge and choke flow boundaries must be strictly managed to avoid safety risks and efficiency losses. The situation is further complicated by the presence of discrete decision variables, such as whether to bypass a compressor or change the boosting direction, which transform the optimization problem into a challenging Mixed-Integer Nonlinear Programming (MINLP) formulation.

Despite advancements in optimization techniques, solving large-scale MINLP problems remains computationally demanding, especially for extensive gas transmission networks. Traditional nonlinear methods or overly simplified linear approximations often fail to capture the system's true complexity or require excessive computation time, limiting their practicality for real-time scheduling applications. As a result, there is a pressing need for efficient, scalable, and accurate algorithms that can deliver high-quality solutions within a reasonable timeframe.

A Sequential Linear Programming (SLP) algorithm is proposed to address the nonlinearities and discrete characteristics inherent in gas network optimization. This method incorporates detailed compressor capacity constraints and operational flexibility, which significantly enhance both solution accuracy and computational efficiency. The algorithm enables rapid optimization for large-scale pipeline networks and provides a practical and robust tool for improving economic performance, operational reliability, and overall system efficiency.

The structure of this paper is as follows: Section 2 reviews literature on natural gas network scheduling and compressor control. Section 3 describes the studied problems and the challenges of integrating compressor capacity constraints into the scheduling process. Section 4 presents the mathematical formulation. Section 5 introduces the proposed solution algorithm. Section 6 discusses numerical experiments and results. Section 7 concludes and outlines future research directions.

## 2. Literature review

### 2.1 Historical context and early models

Optimization of natural gas pipeline operations has been a focus of research for decades, driven by the need to reduce transmission costs and ensure reliable supply amid fluctuating demand. Early approaches relied on linear programming (LP) and mixed-integer linear programming (MILP) to solve simplified versions of the problem.

For example, O'Neill et al. (1979) [1] proposed a linear programming-based method for optimizing transmission network design. The approach identifies capacity

shortages and recommends circuit expansions based on a flow estimation technique that minimizes total circuit length while enhancing network efficiency. Similarly, Hui and Yukikazu (1996) [2] developed a multi-period utility system model based on mixed-integer programming to optimize industrial operations, achieving notable gains in efficiency and cost savings while considering constraints and future demand.

Although these early models provided valuable insights, they lacked the ability to capture the nonlinear behavior inherent in natural gas flow and compression. Gas transport involves complex thermodynamic and fluid dynamic principles, which linear models fail to accurately represent. This limitation has led to the development of more advanced methods that can handle such nonlinearities.

### 2.2 Nonlinear modelling approaches

To overcome the limitations of linear models, researchers developed nonlinear programming (NLP) and mixed-integer nonlinear programming (MINLP) models that more accurately reflect the physical and operational characteristics of natural gas pipelines. These models consider a range of factors, including pressure drops due to friction, changes in gas composition, temperature variations, and the nonlinear relationship between compressor power consumption and pressure ratios.

Wong and Larson (1968) [3] introduced a dynamic programming approach to solve the nonlinear gas transmission network problem. Their model incorporated the nonlinearities in gas flow equations, representing a significant advancement over earlier linear models. Building on this, Edgar et al. (1978) [4] developed a sophisticated algorithm for optimizing the design of gas transmission networks. They used the generalized reduced gradient method to handle nonlinearities and efficiently determine the optimal layout and operation of pipelines and compressors for cost minimization. The method was later applied to gas network operations, addressing objectives such as minimizing fuel consumption and maximizing throughput, while managing the nonlinearities and discrete variables inherent in pipeline optimization (Percell and Ryan, 1987 [5]).

Although these nonlinear models provided a more realistic representation of pipeline operations, they also introduced significant computational challenges. Due to the problem's inherent nonconvexity, traditional optimization techniques often failed to find global optima, which motivated the development of more robust solution strategies.

### 2.3 Mixed-Integer Nonlinear Programming (MINLP) for Pipeline Scheduling

To address the complexities of pipeline scheduling, which include modeling discrete decisions such as compressor activation and changes in flow direction, researchers have developed mixed-integer nonlinear programming (MINLP) models. These models integrate continuous variables (e.g., pressures, flows) with discrete variables (e.g., compressor on/off states) to provide a more comprehensive representation of pipeline operations.

Geißler et al. (2013) [6] introduced a method for solving nonconvex MINLP problems using adaptive refinement of MIP-relaxations, focusing on minimizing energy costs in gas transport networks. This method effectively manages discrete decisions, such as valve and compressor operations. Building on these foundations, more recent studies by Mikolajková et al. (2018) [7] and Burlacu et al. (2019) [8] have further refined these models, incorporating additional physical constraints and operational flexibilities, such as bidirectional flow and multiple compressor configurations.

The inclusion of these additional complexities has significantly increased the difficulty of solving MINLP models. However, advancements in optimization algorithms, including branch-and-bound, branch-and-cut, and decomposition methods, have enhanced the ability to find high-quality solutions within reasonable computational times (Liu et al., 2021 [9]; Zoppei et al., 2022 [10]).

### 2.4 Compressor station optimization

Compressor stations play a vital role in maintaining pressure levels across natural gas pipelines. As a result, their optimization is essential for efficient network operations. Since compressors account for a significant portion of operational costs, optimizing their use can lead to substantial savings.

Early studies approached compressor optimization using simplified models that treated compressors as black boxes with fixed pressure ratio ranges and empirical power consumption estimates (Wong and Larson, 1968 [3]; Sood et al., 1971 [11]). These models failed to capture the complex interactions between compressor capacity and operating conditions such as running speed, inlet pressure, and gas flow rate.

More recent research has developed detailed models that incorporate compressor performance characteristics and interactions within stations. For example, Zhao et al. (2021) [12] proposed a model that accounts for various configurations (series or parallel), pressure ratios, and operational states, thereby providing a more comprehensive understanding of station operations. They also investigated bypass operations—where gas flows through a station without compression—and flow reversal capabilities, both of which enhance the flexibility of pipeline scheduling.

## 2.5  Advanced computational techniques

Given the complex, nonlinear, and nonconvex nature of natural gas pipeline optimization problems, advanced computational techniques have been widely employed to improve solution efficiency and quality. Sequential linear programming (SLP) is one such method. For instance, Chaudry et al. (2008) [13] applied SLP to address a multi-period optimization problem in a combined gas and electricity network.

In addition to SLP, various heuristic and metaheuristic algorithms—such as genetic algorithms (GA), tabu search (TS), and particle swarm optimization (PSO)—have been used to tackle the nonconvexity and discrete decision-making inherent in pipeline scheduling problems (Djebedjian et al., 2008 [14]; Borraz-Sánchez et al., 2005 [15]; Moetamedzadeh et al., 2022 [16]). Although these methods do not guarantee global optimality, they often produce high-quality solutions more efficiently than traditional optimization approaches.

Metaheuristic techniques have also gained considerable traction in broader energy system scheduling problems, which involve nonlinear cost functions, operational constraints, and environmental considerations. For example, Habib et al. (2023) [17] proposed an improved honey bee mating optimization (HBMO) algorithm for economic-emission dispatch, accounting for complex constraints such as valve-point effects, ramp limits, and prohibited operating zones. Shokouhandeh et al. (2022) [18] employed a modified imperialist competitive algorithm (ICA) for optimal microgrid scheduling, incorporating electric vehicle parking lots and distributed generation, and achieved significant cost reductions, outperforming GA and PSO.

To address multi-objective optimization under uncertainty, Kamarposhti et al. (2025) [19] applied NSGA-II for microgrid operation management, achieving effective trade-offs between cost and emissions while incorporating renewable generation variability and load uncertainty. Similarly, Shokouhandeh et al. (2021) [20] introduced a modified gray wolf optimization (MGWO) algorithm for unit commitment under electricity price uncertainty, demonstrating superior performance over conventional GWO and PSO methods.

In the context of distribution systems, Kamarposhti et al. (2021) [21] employed the artificial bee colony (ABC) algorithm to optimize the location and sizing of distributed generators and capacitor banks, thereby reducing power losses and enhancing reliability. Makahleh et al. (2023) [22] optimized the placement and dispatch of energy storage systems for peak shaving using GA and PSO, achieving flattened load curves and enhanced operational stability. Eslami and Kamarposhti (2019) [23] further explored hybrid energy system design by proposing a modified bee algorithm that minimized life cycle costs while maintaining network reliability.

Beyond optimization, fuzzy logic has been applied in protection systems. Rad et al. (2011) [24] developed a fuzzy-based transformer differential protection scheme capable of distinguishing internal faults from non-fault phenomena (e.g., inrush currents and CT saturation), enabling fast and reliable fault detection within half a cycle.

## 2.6  Emerging trends and system-level perspectives

Recent research has extended traditional natural gas pipeline optimization toward clean energy system integration, intelligent operation, and resilience under uncertainty. Peng et al. (2025) [25] proposed a steady-state MINLP model for natural

gas transportation optimization, using a sequential linear programming (SLP) algorithm to improve the efficiency and sustainability of clean energy supply chains. Similarly, Peng et al. (2024) [26] developed a multi-period integrated scheduling framework to manage time-varying demand and operational constraints in complex pipeline systems.

Beyond steady-state optimization, Liu et al. (2025) [27] explored pipeline network design from a cleaner production perspective, emphasizing environmental performance and carbon footprint reduction. Liu et al. (2024) [28] introduced a novel deep reinforcement learning-based optimization framework for dynamic pipeline network decision-making, capturing nonlinear and time-dependent system behaviors.

Emergency and resilience scenarios are also gaining attention. Sun et al. (2024) [29] applied differential evolution to solve emergency scheduling problems within natural gas supply chains, thereby improving robustness during disruption events. Logsdon et al. (2025) [30] focused on pipeline recovery optimization after incidents, using heuristics to accelerate response times and minimize economic losses.

In terms of compressor station operation, Shui et al. (2023) [31] presented a monitoring and optimization method aimed at improving compressor efficiency in large-scale systems. For infrastructure planning, Alavi et al. (2024) [32] incorporated seismic risk into gas pipeline routing through metaheuristic algorithms, enhancing system safety and long-term resilience.

From a mathematical modeling perspective, Adjei et al. (2024) [33] proposed a quadratically constrained MINLP formulation that effectively captures nonlinear flow-pressure relationships and discrete decisions. These diverse contributions underscore a broader shift toward intelligent, sustainable, and adaptive gas network operation strategies.

Recent research also emphasizes the importance of renewable energy integration and intelligent energy management. For instance, microgrid operation and control strategies have been widely reviewed, highlighting their role in reducing costs and enhancing sustainability (Abdelsattar et al., 2023 [34]; Abdelsattar et al., 2024 [35]). In addition, advanced machine learning and deep learning techniques have been applied to improve renewable energy forecasting accuracy (Abdelsattar et al., 2024 [36]; Abdelsattar et al., 2025 [37]), while metaheuristic algorithms such as the African Vultures Optimization Algorithm have been proposed for hybrid system design (Abdelsattar et al., 2024 [38]). These studies illustrate the methodological advances in energy systems optimization; however, they seldom consider compressor capacity constraints in large-scale gas transmission networks, which is the gap addressed in this work.

## 2.7 Our Contributions

This paper addresses critical gaps in natural gas pipeline optimization by introducing practical advancements in both model realism and computational efficiency. Specifically:

(1) We develop a comprehensive compressor modeling framework that captures nonlinear compressor characteristics, including surge and choke flow constraints, thereby significantly enhancing model realism and ensuring better operational safety.

(2) We propose an efficient SLP algorithm, specifically designed to solve large-scale mixed-integer nonlinear programming problems quickly and reliably, effectively overcoming the computational limitations of existing methods.

(3) We validate our approach through real-world experiments on China's largest natural gas pipeline networks, demonstrating its ability to provide near-optimal solutions within practical timeframes, thereby improving both economic efficiency and operational performance.

(4) Problem Description

This chapter addresses practical aspects of natural gas network scheduling, structured into three sections. The first section defines the optimization problem. The second section investigates stable operating conditions for compressors. The third section analyzes how series and parallel compressor configurations affect stability-related operational constraints.

## 3.1 Structure of Gas Network

A long-distance natural gas pipeline network comprises several essential components. Pipeline segments form the primary infrastructure, connecting endpoints and governed by decision variables such as mass flow rate and flow direction, which must satisfy hydraulic constraints related to pressure drops. Control valves, positioned along pipeline segments, reduce gas pressure beyond frictional losses to meet downstream requirements. Their decision variables include flow rate, as well as inlet and outlet pressures. Compressor stations, located within pipeline segments, increase gas pressure to support transmission. Relevant variables are flow rate, rotational speed, compression ratio, and boosting direction, all maintained within stable operating ranges. Gas sources, including fields, storage facilities, liquefied natural gas (LNG) regasification plants, and ports, supply pressurized gas subject to upper and lower supply limits [39]. Demand centres, such as urban and industrial consumers, are defined by consumption constraints and tiered pricing. The decision variable is the quantity of gas sold. Nodes function as junctions between pipeline segments, with pressure as the primary decision variable, constrained within specified bounds.

## 3.2 Modelling of Compression Capacity

A gas compressor is a mechanical device that increases the pressure of a gas to facilitate pipeline transportation. The primary types are reciprocating (piston-type) and centrifugal compressors. As pipeline diameters and flow rates increase, centrifugal compressors are preferred for their operational efficiency and mechanical simplicity. In operation, gas enters the centrifugal compressor axially through a high-speed impeller. Centrifugal force causes the gas to move outward into the diffuser, where the cross-sectional area increases. This transition converts part of the gas's kinetic energy into pressure energy, reducing velocity and increasing pressure. Gas may be recirculated through a return channel to the impeller, enabling multiple compression stages until the desired pressure is achieved.

In gas network optimization, compressor capacity is typically characterized by the compression ratio. This ratio mainly depends on the compressor's rotational speed and inlet flow rate. The relationship between these variables—commonly referred to as the compression ratio curve—is typically derived from experimental data provided by compressor manufacturers. It can be accurately approximated using a high-order polynomial function. In this study, the compression ratio curves of most centrifugal compressors are effectively fitted by high-order polynomials [12]. Specifically, the rotational speed is modeled up to the third order, and the inlet flow rate up to the second order. The specific form of this high-order polynomial is given below.

$$r(f^{in},\ n) = \left(a_1 n^3 + a_2 n^2 + a_2 n + a_4\right) f^{in^2} + \left(b_1 n^3 + b_2 n^2 + b_2 n + b_4\right) f^{in} + \left(c_1 n^3 + c_2 n^2 + c_2 n + c_4\right)$$

In this context, $r(f^{in},\ n)$ denotes the compression ratio function. The coefficients $a_i$, $b_i$, and $c_i$ are obtained through regression fitting. Here, $f^{in}$ represents the inlet flow rate, and $n$ is the compressor's rotational speed.

For centrifugal compressors, at a fixed rotational speed, an increase in inlet flow rate typically reduces the compression ratio, whereas a decrease in inlet flow rate raises it. The inlet flow rate must remain within a specified operating range. Falling below this range may cause a surge, significantly reducing compressor efficiency and service life [40]. Therefore, flow rates below this threshold—referred to as the surge flow rate—are strictly avoided in practice. Conversely, exceeding the upper limit may result in choke, a condition in which the impeller's energy is dissipated in overcoming flow resistance without contributing to pressure gain [40]. Thus, flow rates exceeding this upper limit—termed the choke flow rate—are also prohibited. The relationships between rotational speed and both surge and choke flow rates are modeled using high-order polynomial functions, as illustrated below.

$$f_{min}(n) = d_0 + d_1 n + d_2 n^2 + d_3 n^3$$

$$f_{max}(n) = e_0 + e_1n + e_2n^2 + e_3n^3$$

Here, $d_i$ and $e_i$ are fitting parameters used to characterize the surge and choke boundaries. The warning surge and choke curves are obtained by appropriately shifting the original curves to introduce a safety margin. In addition, the compressor's rotational speed is limited to a specified range, namely $n \in [n_{min}, n_{max}]$. As illustrated in Fig 1, the warning surge and choke curves, together with the minimum and maximum speed limits, define the compressor's stable operating region on the compression ratio surface.

### 3.3 The impact of compressor station layout

Compressor stations in long-distance natural gas pipelines generally include multiple centrifugal compressors arranged in series or parallel. Parallel configurations are used in trunk pipelines to accommodate higher flow rates. Series configurations provide higher compression ratios and are suitable for pipelines with significant elevation changes.

When compressors are configured in parallel, their hydraulic behavior follows a key principle: at a given pressure ratio, the total flow rate equals the sum of individual flow rates [41]. Fig 2 illustrates this principle, where the horizontal axis denotes the inlet flow rate and the vertical axis denotes the pressure ratio. Under constant rotational speed, the combined performance curve of two parallel compressors (I + II) is obtained by summing their individual curves along the flow axis. In this study, all compressors in a parallel configuration are assumed to have identical performance characteristics, including

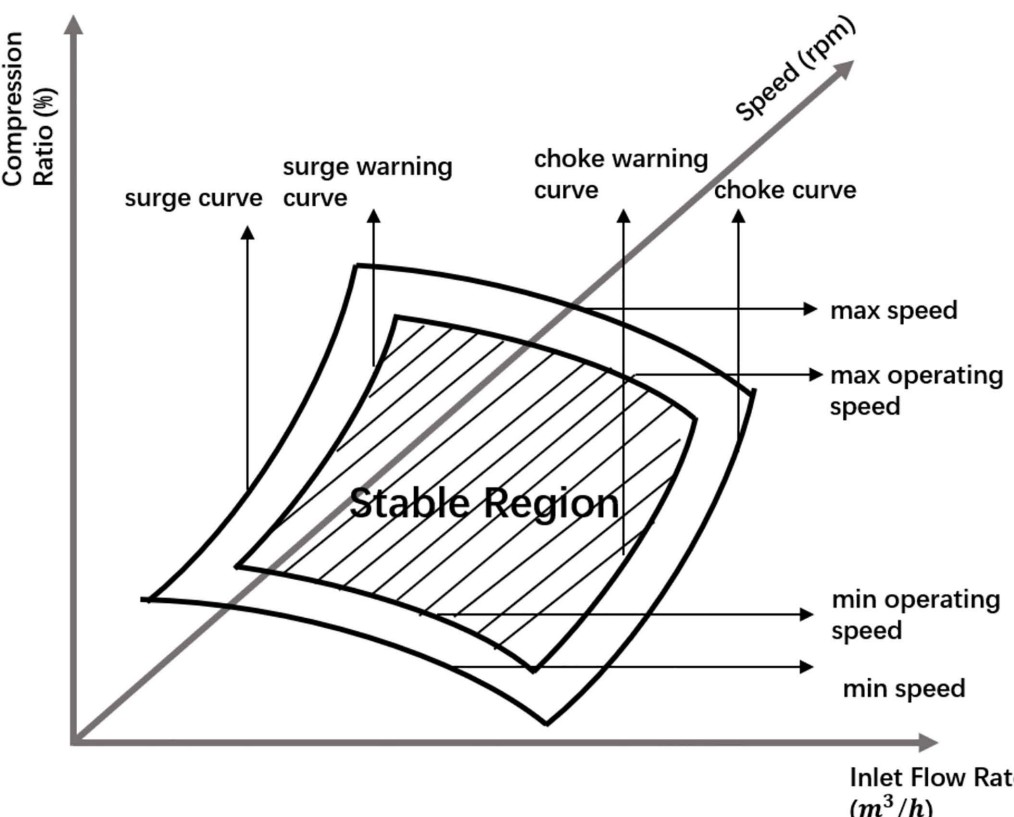

**Fig 1. Stable operating region for compressor.**

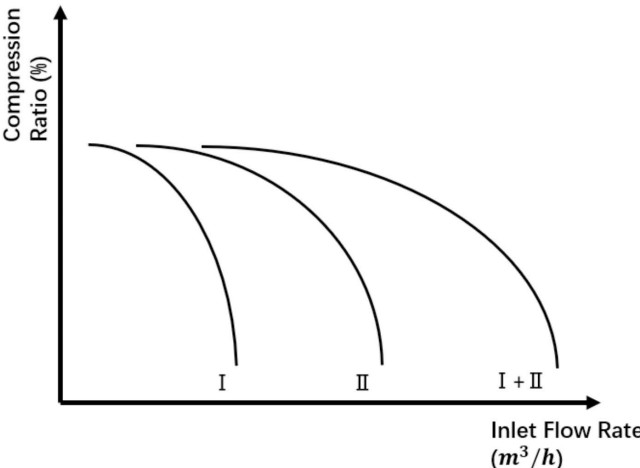

**Fig 2. Compression ratio of compressor station with parallel layout.**

pressure ratio, surge, and choke curves, and the inlet flow is evenly distributed among all units. Based on these assumptions, the aggregate performance curve of the parallel compressor station can be expressed as follows:

Compression Ratio Curve:

$$r(f^{in}, n) = \left( a_1 n^3 + a_2 n^2 + a_2 n + a_4 \right) \frac{f^{in2}}{\tau^2} + \left( b_1 n^3 + b_2 n^2 + b_2 n + b_4 \right) \frac{f^{in}}{\tau} + \left( c_1 n^3 + c_2 n^2 + c_2 n + c_4 \right)$$

Surge Curve:

$$f_{min}(n) = \tau(d_0 + d_1 n + d_2 n^2 + d_3 n^3)$$

Choke Curve:

$$f_{max}(n) = \tau(e_0 + e_1 n + e_2 n^2 + e_3 n^3)$$

Here, $\tau$ denotes the number of compressors arranged in parallel within the station.

When compressors are arranged in series, the overall compression ratio is equal to the product of the individual compression ratios, assuming a constant mass flow rate through each unit [41]. This relationship is illustrated in Fig 3. At a fixed rotational speed, the horizontal and vertical axes correspond to flow rate and compression ratio, respectively. The combined curve for two compressors in series (I*II) is obtained by multiplying their individual compression ratios pointwise along the vertical axis. Although the mass flow rate remains constant across series compressors, changes in pressure and temperature alter the gas's volumetric flow rate and thermodynamic state. These values must be converted using equations of state before applying the compression ratio curves. Given the complexity of individual performance curves, aggregating them into a unified curve for the series configuration is impractical for optimization purposes. Therefore, each compressor is modeled individually in the network, with its respective performance curve and associated decision variables explicitly incorporated into the optimization model.

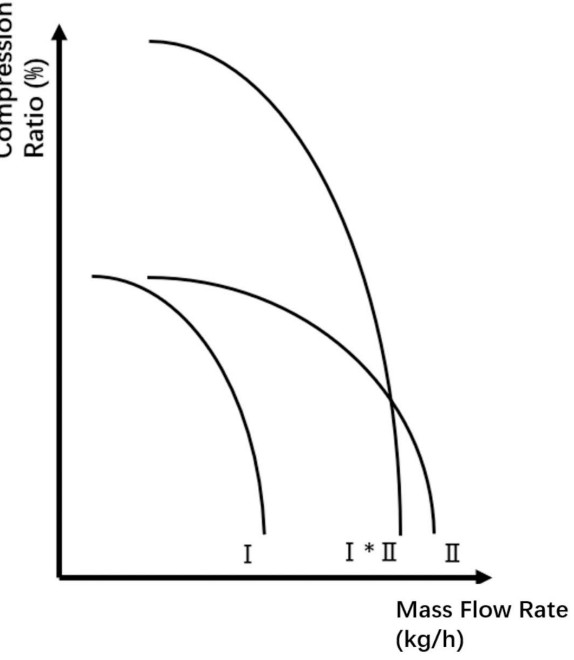

**Fig 3. Compression ratio of Compressor Station with Serial Layout.**

## The mathematical programming model

This chapter presents the formulation of the natural gas network scheduling problem as a nonlinear mixed-integer programming (MINLP) model, referred to as the Nonlinear Original Problem (NOP). The sets, parameters, and variables used in the model are summarized in Table 1–2, and Table 3.

### 4.1 Modelling of Gas Network

The following presents the mathematical formulation of constraints associated with the natural gas pipeline network. Constraint (1) enforces the flow balance at each network node, ensuring that the sum of incoming gas flow and local supply equals the sum of outgoing flow and total demand. Constraints (2) and (3) define the total demand $d_i$ and total supply $o_i$ at node $i$, respectively.

$$\sum_{(j,i)\in O_i} f_{ji}\left(2x_{ji}-1\right) + o_i = d_i + \sum_{(i,j)\in D_i} f_{ij}\left(2x_{ij}-1\right), \forall i \in N \tag{1}$$

$$d_i = \sum_{k\in DM_i} d_{ik}, \forall i \in N \tag{2}$$

$$o_i = \sum_{k\in SP_i} o_{ik}, \forall i \in N \tag{3}$$

Constraint (4) captures the physical relationship between gas pressure and flow in standard transmission pipelines, excluding those equipped with compressor stations or control valves. The binary variable $x_{ij}$ indicates whether the flow

**Table 1. Sets used for modelling the gas network optimization problem.**

| Set | Description |
|---|---|
| $N$ | Set of all network nodes |
| $SP_i$ | Set of gas sources belonging to node i |
| $DM_i$ | Set of demand centres belonging to node i |
| $ST_k$ | Set of price tiers belonging to demand k |
| $PI$ | Set of network arcs representing standard pipeline segments (excluding booster stations and control valves). |
| $CP$ | Set of network arcs representing compressor stations. |
| $VP$ | Set of network arcs representing control valves. |
| $O_i$ | Set of pipelines starting from node i |
| $D_i$ | Set of pipelines ending from node i |

**Table 2. Parameters used for modelling the gas network optimization problem.**

| Parameters | Description |
|---|---|
| $F_{ij}^{ub}$ | The upper bound flow rate of the pipeline segment from node i to node j. |
| $d_{ik}^{lb}$ | The lower bound of the k-th demand at node i |
| $d_{ik}^{ub}$ | The upper bound of the k-th demand at node i |
| $o_{ik}^{lb}$ | The supply lower bound of the k-th gas source at node i |
| $o_{ik}^{ub}$ | The supply upper bound of the k-th gas source at node i |
| $p_i^{lb}$ | The lower pressure bound at node i |
| $p_i^{ub}$ | The upper pressure bound at node i |
| $c_{ik}^{gj}$ | The supply price of the k-th gas source at node y |
| $c_{ij}^{ys}$ | The transportation unit price from node i to node j |
| $M_{ik}$ | The number price tiers in the k-th demand at node i. |
| $q_{ikm}$ | The maximum demand volume of the m-th step of the k-th demand at node i |
| $p_{ikm}$ | The selling price of the m-th step of the k-th demand at node i |
| $R_{ikm}$ | The total sales amount for the first $m-1$ steps when the usage reaches the m-th step |
| $Q_{ikm}$ | The total quantity sold for the first $m-1$ steps when the usage reaches the m-th step |
| $\gamma, \theta, \beta$ | $\gamma, \theta, \beta$ are parameters derived from the hydraulic constraint with elevation differences. $$\gamma = \frac{\lambda Z \Delta_{\bar{s}} TL}{D^5}$$ $$\theta = \left( \frac{a}{2L} \sum_{i=1}^{n} (h_i + h_{i-1}) L_i \right)$$ $$\beta = a \cdot \Delta h$$ |
| $\in$ | State conversion factor derived from the equation of state. $$\in = \frac{Z_{in} T_{in} p_0}{Z_0 T_0}$$ |
| $n_{min}$ | The minimum operating rotational speed of the compressor station |
| $n_{max}$ | The maximum operating rotational speed of the compressor station |
| $M$ | Big M |
| $a_i, \ b_i, c_i$ | Regression coefficients of the compression ratio function |
| $d_i$ | Regression coefficients of the surge curve |
| $e_i$ | Regression coefficients of the choke curve |
| $\tau$ | Number of compressors arranged in parallel within the compressor station |
|  | The neighbourhood radius for flow variables in SLP iterations |
| $\varepsilon_{pressure}$ | The neighbourhood radius for pressure variables in SLP iterations |

**Table 3. Variables for modelling the gas network optimization problem.**

| Variables | Domain | Description |
|---|---|---|
| $p_i$ | $\mathbb{R}_{\geq 0}$ | The pressure value at node i |
| $f_{ij}$ | $\mathbb{R}_{\geq 0}$ | The absolute value of the flow rate via the pipeline between node i and j under standard conditions, independent of direction |
| $x_{ij}$ | $\{0,1\}$ | The flow direction of the pipeline between node i and j. It is 1 if the flow is from i to j, and 0 otherwise |
| $y_{ij}$ | $\{0,1\}$ | The bypass decision of compressor stations between node i and j. It is 1 if the gas compressor station is bypassed, and 0 otherwise |
| $d_{ik}$ | $\mathbb{R}_{\geq 0}$ | The satisfied volume for the k-th demand at node i |
| $d_i$ | $\mathbb{R}_{\geq 0}$ | The total demand volume at node i. |
| $o_{ik}$ | $\mathbb{R}_{\geq 0}$ | The supply volume from the k-th source at node i |
| $o_i$ | $\mathbb{R}_{\geq 0}$ | The total supply volume at node i. |
| $v_{ikm}$ | $\{0,1\}$ | Binary variable indicating whether $d_{ik}$ in the volume interval of the m-th price step |
| $u_{ikm}$ | $\mathbb{R}_{\geq 0}$ | The volume that $d_{ik}$ exceeds the volume lower bound of the m-th price step, if $d_{ik}$ in the volume interval of the m-th price step, and 0 otherwise |
| $f_{ij}^{in}$ | $\mathbb{R}_{\geq 0}$ | The flow rate through the compressor station between nodes i and j under inlet conditions |
| $f_{ij}^{max-in}$ | $\mathbb{R}_{\geq 0}$ | The maximum flow rate of the compressor station between nodes i and j under inlet conditions, known as the choke warning flow rate |
| $f_{ij}^{min-in}$ | $\mathbb{R}_{\geq 0}$ | The minimum flow rate of the compressor station between nodes i and j under inlet conditions, known as the surge warning flow rate |
| $p_{ij}^{in}$ | $\mathbb{R}_{\geq 0}$ | The inlet pressure of the compressor station between nodes i and j |
| $r_{ij}$ | $\mathbb{R}_{\geq 0}$ | The compression ratio of the compressor station between nodes i and j |
| $n_{ij}$ | $\mathbb{R}_{\geq 0}$ | The rotational speed of the compressor station between nodes i and j |
| $p_i'$ | $\mathbb{R}_{\geq 0}$ | The pressure increases at the compressor station relative to the inlet pressure $p_i$ if the compression direction is from i to j and there is no bypass; otherwise, it is 0. |
| $\xi_{ij}^+$ | $\mathbb{R}_{\geq 0}$ | The positive slack variable associated with the hydraulic constraint of the pipeline segment from node $i$ to node $j$ |
| $\xi_{ij}^-$ | $\mathbb{R}_{\geq 0}$ | The negative slack variable associated with the hydraulic constraint of the pipeline segment from node $i$ to node $j$ |
| $\psi_{ij}^+$ | $\mathbb{R}_{\geq 0}$ | The positive slack variable associated with the pressure balance constraint of the compressor station between node $i$ and node $j$ |
| $\psi_{ij}^-$ | $\mathbb{R}_{\geq 0}$ | The negative slack variable associated with the pressure balance constraint of the compressor station between node $i$ and node $j$ |
| $\zeta_{ij}^+$ | $\mathbb{R}_{\geq 0}$ | The positive slack variable associated with the flow range constraint of the compressor station between node $i$ and node $j$ |
| $\zeta_{ij}^-$ | $\mathbb{R}_{\geq 0}$ | The negative slack variable associated with the flow range constraint of the compressor station between node $i$ and node $j$ |
| $\gamma_{ik}$ | $\mathbb{R}_{\geq 0}$ | The slack variable associated with the lower bound constraint of the $k$-th demand at node $i$ |

direction is from node $i$ to node $j$. When $x_{ij} = 0$ (i.e., the flow direction is from $j$ to $i$), the expression $(2x_{ij} - 1)$ evaluates to -1, thereby reversing the pressure–flow relationship in the equation.

$$\gamma \left(1 + \theta + \left(x_{ij} - 1\right)\beta\right) f_{ij}^2 = \left(2x_{ij} - 1 - \left(x_{ij} - 1\right)\beta\right) p_i^2 - \left(2x_{ij} - 1 + \beta x_{ij}\right) p_j^2, \ \forall (i,j) \in PI \tag{4}$$

The coefficients in Equation (4) are defined as follows:

$$\gamma = \frac{\lambda Z \Delta^* TL}{D^5}$$

$$\theta = \left( \frac{a}{2L} \sum_{l=1}^{n} \left( h_l + h_{l-1} \right) L_l \right)$$

$$\beta = a \cdot \Delta h$$

Where:

$T$ is the temperature,

$D$ is the inner diameter of the pipeline segment

$L$ is the length of the pipeline segment,

$\Delta^*$ is the relative density,

$\lambda$ is the hydraulic friction coefficient,

$Z$ is the compressibility factor,

$h_l$ are the elevations at the endpoints of pipeline sub-segment $l$, introduced to capture topographic variability,

$L_l$ are the lengths of pipeline sub-segment $l$,

$\Delta h$ is the elevation difference between node $i$ and node $j$, and

$\alpha$ is the elevation coefficient, representing the contribution of elevation differences to pressure losses in the pipeline.

Constraints (5) and (6) describe the physical relationship between gas pressure and flow in transmission pipelines equipped with control valves. Specifically, when the transmission direction coincides with the pressure relief direction of the control valve (i.e., $x_{ij} = 1$), the outlet pressure is allowed to decrease to any value above the minimum pressure limit. Otherwise, the pressure–flow relationship must conform to the standard pipeline constraints.

$$\gamma \left( 1 + \theta + \left( x_{ij} - 1 \right) \beta \right) f_{ij}^2 \leq \left( 2x_{ij} - 1 - \left( x_{ij} - 1 \right) \beta \right) p_i^2 - \left( 2x_{ij} - 1 + \beta x_{ij} \right) p_j^2, \forall (i,j) \in VP \tag{5}$$

$$\gamma \left( 1 + \theta + \left( x_{ij} - 1 \right) \beta \right) f_{ij}^2 \geq (x_{ij} - 1)((1 - \beta)p_i^2 - p_j^2), \forall (i,j) \in VP \tag{6}$$

This section presents the constraints related to gas sources and demand. In the case study, the sales price of natural gas at demand points is modelled as a step function of the demand volume. Accordingly, constraints are introduced to capture the price-tier assignment based on the quantity demanded. Constraint (7) ensures that the demand at each natural gas demand point falls exclusively within a single price tier.

$$\sum_{m \in ST_k} v_{ikm} = 1, \ \forall i \in N, \ \forall k \in DM_i \tag{7}$$

Constraint (8) ensures that the variable $u_{ikm}$ takes a nonzero value only when the total sales volume $d_{ik}$ falls within a specific pricing tier; otherwise, it must be zero. Specifically, if the sales volume falls within tier $m$, then $v_{ikm}$ is assigned a value of 1, constraining $u_{ikm}$ to be less than or equal to the upper limit of that tier, $q_{ikm}$. Conversely, if the sales volume does not fall within tier $m$, then $v_{ikm} = 0$, thereby enforcing $u_{ikm} = 0$. Constraint (9) ensures that if $d_{ik}$ falls within tier $m$, then it equals the cumulative sales volume of tiers 1 through $m - 1$, denoted $Q_{ikm}$, plus the remaining volume within tier $m$, $u_{ikm}$.

$$u_{ikm} \leq v_{ikm} q_{ikm}, \ \forall i \in N, \ \forall k \in DM_i, \ \forall m \in ST_k \tag{8}$$

$$\sum_{m \in ST_k} (v_{ikm} Q_{ikm} + u_{ikm}) = d_{ik}, \ \forall i \in N, \ \forall k \in DM_i \tag{9}$$

 

Constraints (10), (11), (12), and (13) impose bound constraints on the demand, supply, pressure, and flow rate variables, respectively.

$$d_{ik}^{lb} \le d_{ij} \le d_{ik}^{ub}, \ \forall i \in N, \ \forall k \in DM_i \tag{10}$$

$$o_{ik}^{lb} \le o_{ij} \le o_{ik}^{ub}, \ \forall i \in N, \ \forall k \in SP_i \tag{11}$$

$$p_i^{lb} \le p_i \le p_i^{ub}, \ \forall i \in N \tag{12}$$

$$f_{ij} \le F_{ij}^{ub}, \ \forall (i,j) \in VP \cup PI \tag{13}$$

### 4.2 Modelling of compressor stations

The following presents the formulation of constraints associated with the compression capacity of compressor stations. Constraint (14) describes the nonlinear relationship among the compression ratio, the compressor station's rotational speed, and the inlet flow rate. Let $r_{ij}(f^{in}, n)$ denote the compression ratio curve of the compressor station between nodes $i$ and $j$, as a function of the inlet flow rate and rotational speed. Constraint (15) defines the conversion between standard and inlet condition flow rates, where $\in$ denotes the conversion factor.

$$r_{ij} = r_{ij}\left(f_{ij}^{in}, \ n_{ij}\right), \forall (i,j) \in CP \tag{14}$$

$$f_{ij}^{in} = \in \frac{f_{ij}}{p_{ij}^{in}}, \forall (i,j) \in CP \tag{15}$$

The following four sets of constraints govern the pressure assignments at both ends of the compressor station based on discrete operational decisions, such as flow direction and bypass status. Constraint (16) characterises the pressure balance between the inlet and outlet of the compressor station during the compression process. When the flow direction is from node $i$ to $j$, Constraints (17) and (18) assign the pressure increase to $p_i'$, while $p_j'$ is set to zero; otherwise, the pressure increase is assigned to $p_j'$, and $p_i'$ is zero. If the compressor station is bypassed, Constraint (19) enforces zero pressure increase at both ends.

$$p_i + p_i' = p_j + p_j', \forall (i,j) \in CP \tag{16}$$

$$p_i' \le r_{ij}p_i - p_i \ , \ \ p_j' \le r_{ij}p_j - p_j, \ \ \forall (i,j) \in CP \tag{17}$$

$$p_i' \le M \cdot x_{ij} \ , \ \ p_j' \le M\left(1 - x_{ij}\right), \ \ \forall (i,j) \in CP \tag{18}$$

$$p_i' \le M\left(1 - y_{ij}\right) \ , \ \ p_j' \le M\left(1 - y_{ij}\right), \ \ \forall (i,j) \in CP \tag{19}$$

Constraints (20) and (21) assign compressor inlet pressure values according to the specified flow direction variables.

$$p_{ij}^{in} \leq p_i , \quad p_{ij}^{in} \leq p_j, \quad \forall (i,j) \in CP \tag{20}$$

$$p_{ij}^{in} \geq p_i - M\left(1 - x_{ij}\right) , \quad p_{ij}^{in} \geq p_j - M \cdot x_{ij} , \quad \forall (i,j) \in CP \tag{21}$$

Let $f_{ij}^{min}(n)$ and $f_{ij}^{max}(n)$ denote the warning surge and choke flow rate curves of the compressor station between nodes $i$ and $j$. The following two sets of constraints define the relationship between the surge and choke warning flow rates—$f_{ij}^{min-in}$ and $f_{ij}^{max-in}$, respectively—and the operating rotational speed of the compressor station.

$$f_{ij}^{min-in} = f_{ij}^{min}(n_{ij}), \forall (i,j) \in CP \tag{22}$$

$$f_{ij}^{max-in} = f_{ij}^{max}(n_{ij}), \forall (i,j) \in CP \tag{23}$$

Constraints (24) and (25) are used to ensure that the compressor station's key operational parameters—flow rate and rotational speed—remain within their stable operating ranges

$$f_{ij}^{min-in} \leq f_{ij}^{in} \leq f_{ij}^{max-in}, \forall (i,j) \in CP \tag{24}$$

$$n_{min} \leq n_{ij} \leq n_{max}, \forall (i,j) \in CP \tag{25}$$

## 4.3 The objective function

The objective of this study is to maximize total network profit. This objective comprises three components: sales revenue at each demand center, procurement cost at the gas source, and total transportation cost, calculated per pipeline segment.

$$max \left( \sum_{\substack{i \in N \\ k \in DM_i \\ m \in ST_k}} \left( v_{ikm}R_{ikm} + u_{ikm}p_{ikm} \right) - \sum_{\substack{i \in N \\ k \in SP_i}} o_{ik}c_{ik}^{gj} - \sum_{(i,j) \in PI \cup CP} f_{ij}c_{ij}^{ys} \right) \tag{26}$$

## 5. Solution algorithm

### 5.1 Sequential linear programming

Natural gas network scheduling is characterized by detailed compressor capacity constraints, numerous nonlinear relationships, and complex discrete decisions, resulting in a challenging nonlinear mixed-integer programming (MINLP) problem. In large-scale networks, conventional nonlinear programming methods, such as piecewise function approximations, frequently fail to provide solutions within a practical timeframe. To address this limitation, Sequential Linear Programming

(SLP) is employed. SLP linearizes nonlinear functions using first-order Taylor expansions, transforming the original problem into a linear approximation problem (LAP). The LAP can be solved using standard mixed-integer programming techniques to obtain locally optimal solutions. The formulation of LAP based on the original nonlinear optimization problem (NOP) is presented below.

**Objective** (26)

**S.t**

**Constraint** (1)(2)(3)(7)(8)(9)(10)(11)(12)(13)(16)
(17)(18)(19)(20)(21)(24)(25)

$$\gamma \left(1 + \theta + \left(x_{ij} - 1\right) \beta\right) \left(f_{ij,0}^2 + 2f_{ij,0} \left(f_{ij} - f_{ij,0}\right)\right) = \left(2x_{ij} - 1 - \left(x_{ij} - 1\right) \beta\right) \left(p_{i,0}^2 + 2p_{i,0} \left(p_i - p_{i,0}\right)\right)$$
$$- \left(2x_{ij} - 1 + \beta x_{ij}\right) \left(p_{j,0}^2 + 2p_{j,0} \left(p_j - p_{j,0}\right)\right), \forall (i,j) \in PI \tag{27}$$

$$\gamma \left(1 + \theta + \left(x_{ij} - 1\right) \beta\right) \left(f_{ij,0}^2 + 2f_{ij,0} \left(f_{ij} - f_{ij,0}\right)\right) \leq \left(2x_{ij} - 1 - \left(x_{ij} - 1\right) \beta\right) \left(p_{i,0}^2 + 2p_{i,0} \left(p_i - p_{i,0}\right)\right)$$
$$- \left(2x_{ij} - 1 + \beta x_{ij}\right) \left(p_{j,0}^2 + 2p_{j,0} \left(p_j - p_{j,0}\right)\right), \forall (i,j) \in VP \tag{28}$$

$$\gamma \left(1 + \theta + \left(x_{ij} - 1\right) \beta\right) \left(f_{ij,0}^2 + 2f_{ij,0} \left(f_{ij} - f_{ij,0}\right)\right) \geq \left(x_{ij} - 1\right) \left(\left(1 - \beta\right) \left(p_{i,0}^2 + 2p_{i,0} \left(p_i - p_{i,0}\right)\right)\right.$$
$$\left. - \left(p_{j,0}^2 + 2p_{j,0} \left(p_j - p_{j,0}\right)\right)\right), \quad \forall (i,j) \in VP \tag{29}$$

$$r_{ij} = r_{ij} \left(f_{ij,0}^{in}, n_{ij,0}\right) + \frac{\partial r_{ij}}{\partial f^{in}} \left(f_{ij,0}^{in}, n_{ij,0}\right) \left(f_{ij}^{in} - f_{ij,0}^{in}\right) + \frac{\partial r_{ij}}{\partial n} \left(f_{ij,0}^{in}, n_{ij,0}\right) \left(n_{ij} - n_{ij,0}\right), \forall (i,j) \in CP \tag{30}$$

$$f_{ij}^{in} = \theta \left[\frac{f_{ij,0}}{p_{ij,0}^{in}} + \frac{1}{p_{ij,0}^{in}} \left(f_{ij} - f_{ij,0}\right) - \frac{f_{ij,0}}{p_{ij,0}^{in\,2}} \left(p_{ij,0}^{in} - p_{ij}^{in}\right)\right] \tag{31}$$

$$f_{ij}^{min-in} = f_{ij}^{min} \left(n_{ij,0}\right) + f_{ij}^{min\,\prime} \left(n_{ij,0}\right) \left(n_{ij} - n_{ij,0}\right), \forall (i,j) \in CP \tag{32}$$

$$f_{ij}^{max-in} = f_{ij}^{max} \left(n_{ij,0}\right) + f_{ij}^{max\,\prime} \left(n_{ij,0}\right) \left(n_{ij} - n_{ij,0}\right), \forall (i,j) \in CP \tag{33}$$

$$-\varepsilon_{flow} \leq f_{ij} - f_{ij,0} \leq \varepsilon_{flow}, \ \forall (i,j) \in VP \cup PI \tag{34}$$

$$-\varepsilon_{pressure} + \left(y_{ij,0} - y_{ij}\right) p_j' \leq p_i - p_{i,0} \leq \varepsilon_{pressure} + \left(y_{ij,0} - y_{ij}\right) p_j',$$

$$-\varepsilon_{pressure} + \left(y_{ij,0} - y_{ij}\right) p_i' \leq p_j - p_{j,0} \leq \varepsilon_{pressure} + \left(y_{ij,0} - y_{ij}\right) p_i', \quad \forall (i,j) \in CP \tag{35}$$

$$-\varepsilon_{pressure} \leq p_i - p_{i,0} \leq \varepsilon_{pressure}, \ \forall i \in N \tag{36}$$

Constraints (27)–(33) represent the linear approximations of constraints(4), (5), (6), (14), (15), (22), and (23), respectively. Here, $f_{ij,0}$, $p_{i,0}$, $n_{ij,0}$, $y_{ij,0}$ and $p_{ij,0}^{in}$ denote the base point values of the corresponding variables $f_{ij}$, $p_i$, $n_{ij}$, $y_{ij}$ and $p_{ij}^{in}$, respectively. Constraints (34)–(36) limit the linearized variables to a neighbourhood around the base point to mitigate approximation errors. In particular, constraint (35) addresses the sensitivity of outlet pressure to changes in the compressor station's bypass variable, which can cause abrupt pressure variations. Directly constraining the pressure variable near the base point may inhibit changes in the bypass decision, potentially leading to inferior solutions. To alleviate this issue, an adjustment term $(y_{ij,0} - y_{ij}) p_i'$ is introduced into constraint (35), allowing dynamic adaptation of the pressure bounds in response to bypass status and thereby preserving solution quality.

In sequential linear programming (SLP), the process begins by identifying an optimal solution within a predefined neighbourhood of the current base point. This solution is then used to update the linear approximation and serves as the new base point for the next iteration. The procedure is repeated, and the neighbourhood radius is gradually reduced when the improvement in the objective value between successive iterations becomes sufficiently small. Once the radius shrinks below a certain threshold, the algorithm is considered to have converged to a local optimum of the original nonlinear problem.

## 5.2 Finding initial base point

Prior to applying linear approximation, it is necessary to identify a feasible solution that satisfies all original constraints to serve as the initial base point. Accordingly, a four-phase solution procedure is developed. The first three phases sequentially address feasibility for hydraulic constraints, compressor capacity constraints, and minimum demand constraints. The final phase focuses on optimizing the overall objective. In Phase One, all three types of constraints are relaxed and slack variables are introduced. The objective is to minimize violations associated with the hydraulic constraints. The slack problem for Phase One (hereafter referred to as LAP-S1) is formulated as follows.

**Objective min** $\sum_{(i,j) \in PI} (\xi_{ij}^- + \xi_{ij}^+)$
**S.t**
**Constraint** (1)(2)(3)(7)(8)(9) (11)(12)(13)(17)(18)
(19)(20)(21) (25)(28)(29)(30)(31)(32)(33)(34)(35)(36)

$$\gamma \left(1 + \theta + (x_{ij} - 1) \beta\right) \left(f_{ij,0}^2 + 2f_{ij,0} (f_{ij} - f_{ij,0})\right) = (2x_{ij} - 1 - (x_{ij} - 1) \beta) \left(p_{i,0}^2 + 2p_{i,0} (p_i - p_{i,0})\right)$$
$$- (2x_{ij} - 1 + \beta x_{ij}) \left(p_{j,0}^2 + 2p_{j,0} (p_j - p_{j,0})\right) + \xi_{ij}^- - \xi_{ij}^+, \forall (i,j) \in PI \tag{37}$$

$$p_i + p_i' = p_j + p_j' + \psi_{ij}^- - \psi_{ij}^+, \forall (i,j) \in CP \tag{38}$$

$$f_{ij}^{min-in} \leq f_{ij}^{in} + \zeta_{ij}^- - \zeta_{ij}^+ \leq f_{ij}^{max-in}, \forall (i,j) \in CP \tag{39}$$

$$d_{ik}^{lb} \leq d_{ik} + \gamma_{ik} \leq d_{ik}^{ub}, \ \forall i \in N, \ \forall k \in DM_i \tag{40}$$

In the Phase One slack problem, slack variables $\xi_{ij}^-$ and $\xi_{ij}^+$ are introduced into the hydraulic constraints (27) and subsequently minimized as the objective. To relax compressor capacity constraints, slack variables $\psi_{ij}^-$ and $\psi_{ij}^+$ are added to the

pressure balance constraints (16) that link inlet and outlet pressures, allowing them to deviate from the compression ratio. Similarly, $\zeta_{ij}^-$ and $\zeta_{ij}^+$ are introduced into the flow range constraints (24) to relax the feasible operating range. Lastly, $\gamma_{ik}$ is included in the minimum demand constraint (10) to soften demand satisfaction requirements.

The process begins by setting the base point at the lower bound for all variables. An optimal solution is then iteratively searched within a neighborhood of this base point, which is updated at each iteration. As the objective value approaches zero, the neighborhood radius is gradually reduced. This process continues until the radius becomes sufficiently small, resulting in a base point that satisfies constraint (27). If the objective value fails to reach zero, it indicates that the original nonlinear problem is infeasible.

In Phase Two, the slack variables associated with the hydraulic constraints are removed, and the solution obtained from Phase One serves as the new initial base point. The objective is redefined to minimize the slack variables corresponding to the compressor capacity constraints, yielding a solution that satisfies these conditions. The slack problem for Phase Two (hereafter referred to as LAP-S2) is formulated as follows:

**Objective min** $\sum_{(i,j) \in CP} (\psi_{ij}^- + \psi_{ij}^+ + \zeta_{ij}^- + \zeta_{ij}^+)$

**S.t**

**Constraint** (1)(2)(3)(7)(8)(9) (11)(12)(13)(17)(18) (19)(20)(21)(25)(27)(28)(29)(30)(31)(32)(33)(34)(35) (36)

$$p_i + p_i' = p_j + p_j' + \psi_{ij}^- - \psi_{ij}^+, \forall(i,j) \in CP \tag{41}$$

$$f_{ij}^{min-in} \leq f_{ij}^{in} + \zeta_{ij}^- - \zeta_{ij}^+ \leq f_{ij}^{max-in}, \forall(i,j) \in CP \tag{42}$$

$$d_{ik}^{lb} \leq d_{ik} + \gamma_{ik} \leq d_{ik}^{ub}, \ \forall i \in N, \ \forall k \in DM_i \tag{43}$$

A similar procedure is followed in Phase Two. Once the objective function reaches zero and the neighborhood radius is sufficiently reduced, a base point satisfying constraints (16) and (24) is obtained. If convergence is not achieved, it suggests that the original nonlinear problem is infeasible.

In Phase Three, the slack variables associated with these two constraints are removed, and the solution obtained from Phase Two is used as the new initial base point. The objective is to find a solution that satisfies the minimum demand constraints. The slack problem for Phase Three (hereafter referred to as LAP-S3) is formulated as follows:

**Objective min** $\sum_{i \in N, k \in DM_i} \gamma_{ik}$

**S.t**

**Constraint** (1)(2)(3)(7)(8)(9) (11)(12)(13)(16)(17) (18)(19)(20)(21)(24)(25)(27)(28)(29)(30)(31)(32)(33) (34)(35)(36)

$$d_{ik}^{lb} \leq d_{ik} + \gamma_{ik} \leq d_{ik}^{ub}, \ \forall i \in N, \ \forall k \in DM_i \tag{44}$$

The same procedure is applied, with termination determined by two criteria: the objective improvement is sufficiently small, and the neighborhood radius is adequately reduced. If the optimal objective value reaches zero, a base point satisfying constraint (10) is identified. Otherwise, the minimum demand constraint is considered infeasible and should be adjusted.

In Phase Four, the linear approximation problem is solved iteratively using termination criteria similar to those in LAP-S3. This final phase yields a locally optimal solution to the original nonlinear problem.

## Numerical experiment

This study presents a case analysis based on China's largest long-distance natural gas pipeline network. Spanning most of the country, the network extends approximately 37,000 kilometers and includes around 650 nodes, 670 pipelines, 119 compressor stations, and 65 control valves. To facilitate the evaluation of the proposed model and algorithms, the network is divided into six test cases of varying sizes and configurations. Each test case features a distinct number of pipeline segments, nodes, and compressor stations, as summarized in Table 4.

The experimental analysis focuses on two primary objectives: (1) assessing the nonlinear approximation errors of key variables to verify algorithmic accuracy, and (2) validating the model's effectiveness by comparing results against those obtained using piecewise linear function-based methods. All experiments were conducted on a system equipped with an AMD Ryzen 5 4600U CPU and 16 GB of RAM. The algorithms were implemented in Python 3.7, with GUROBI 11 employed as the MIP solver.

In the numerical experiments, the proposed algorithm is tested on gas networks of varying scales, derived from different segments of the national pipeline system. The selected cases represent key regions, including Northeast China, Northwest China, Southwest China, Southern China, Northern China, and the entire national network. The testing aims to achieve two primary objectives: minimizing violations of demand lower bounds and maximizing sales profit.

For each case, both the computational time and objective function values are recorded. These results are then compared against those obtained using a MINLP solver based on piecewise linear approximations—specifically, the non-convex MINLP solver integrated into GUROBI. This comparison aims to assess the proposed algorithm's computational efficiency and the quality of the solutions it generates.

### 6.1 Improvement process

To comprehensively evaluate the performance and convergence behavior of the proposed SLP algorithm, we analyzed its iterative optimization process using the national-scale natural gas pipeline network as a representative case. Fig 4 illustrates the evolution of objective values across optimization phases over computational time. The horizontal axis represents the elapsed CPU time, while the vertical axis indicates the corresponding objective values at each iteration.

In the initial phase (LAP-S1), which focuses on satisfying hydraulic feasibility, the algorithm typically converges rapidly, producing a feasible solution with negligible constraint violations within the first iteration. As a result, only the improvement trajectories for the subsequent three phases (LAP-S2, LAP-S3, and LAP) are depicted.

The second phase (LAP-S2) targets compressor capacity constraints and also demonstrates rapid convergence. Feasible compressor operating points that satisfy complex nonlinear constraints are generally identified within a few seconds.

**Table 4. Scale of test cases.**

| Instance | Num of Pipelines | Num of Nodes | Num of Compressors |
|---|---|---|---|
| Northeast China | 56 | 64 | 4 |
| Northwest China | 46 | 46 | 17 |
| Southwest China | 88 | 97 | 8 |
| Southern China | 258 | 262 | 23 |
| Northern China | 402 | 390 | 95 |
| Entire China | 655 | 642 | 118 |

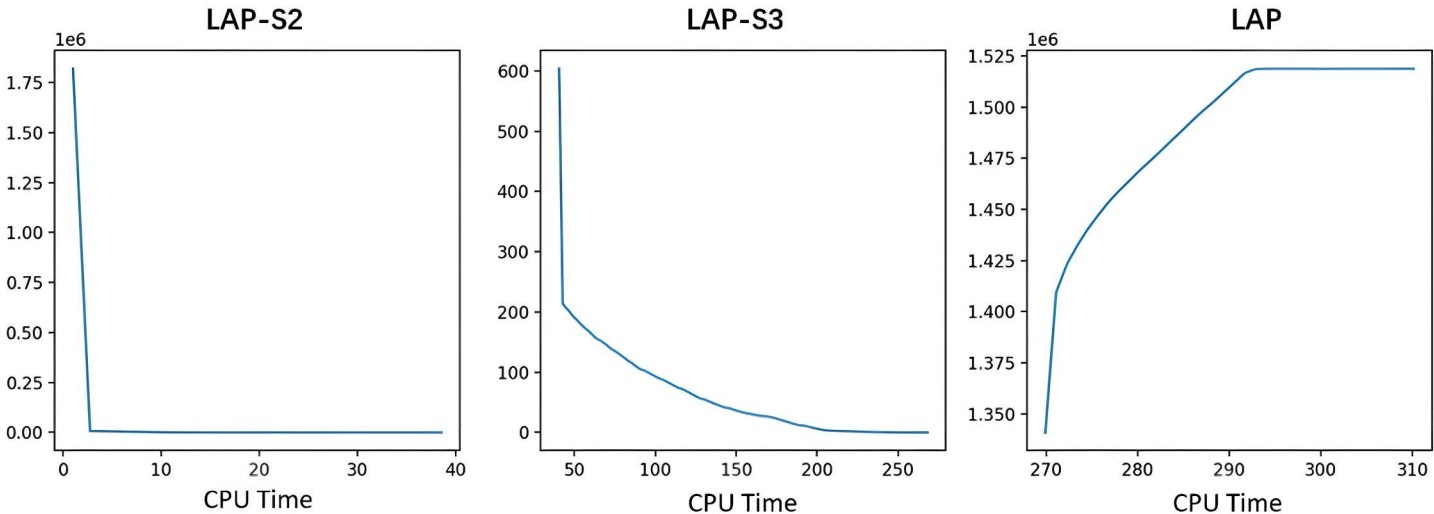

**Fig 4. Improvement curves for the "Entire China" instance.**

This swift progress highlights the effectiveness of the linear approximation strategy in navigating the nonlinear feasible region associated with compressor operations.

In contrast, the third phase (LAP-S3), which enforces minimum demand constraints, converges more slowly due to the increased complexity of balancing supply and demand across the network. This phase typically requires approximately 200 seconds of iterative adjustments to fully satisfy all demand lower bounds. The extended computation time reflects the inherent challenge of coordinating supply across numerous nodes in a large-scale system.

The final phase (LAP), aimed at maximizing sales profit, exhibits stable and consistent convergence. The objective improves rapidly in the early iterations and gradually stabilizes, reaching an optimal or near-optimal solution within approximately 290 seconds. This steady convergence confirms the algorithm's ability to efficiently explore the complex solution space and deliver economically favorable scheduling outcomes.

The four-phase sequential optimization process consistently produces high-quality locally optimal solutions within approximately 300 seconds for the largest national-scale network. Smaller-scale cases require substantially less computational time, demonstrating the scalability, efficiency, and practical applicability of the proposed SLP method for real-world gas pipeline scheduling.

### 6.2 Analysis of non-linear error

Given that the SLP algorithm relies on linear approximations, it is essential to assess the deviation introduced by this simplification. Table 5 reports the maximum and average errors of three critical compressor-related constraints—compression ratio, surge flow rate, and choke flow rate—under different neighborhood radii. Two representative networks, the national-scale "Entire China" system and the regional "Northwest China" system, were analyzed.

Across both cases, the results reveal a clear monotonic trend: as the neighborhood radius decreases, nonlinear errors drop rapidly, often by one to two orders of magnitude. For example, in the "Entire China" instance, reducing the radius from $5.0 \times 10^{-1}$ to $5.0 \times 10^{-2}$ lowers the maximum error of the compression ratio from $1.6 \times 10^{-2}$ to $1.85 \times 10^{-4}$, a reduction of nearly two orders of magnitude. Similar improvements are observed for the surge and choke flow constraints. This confirms that smaller radii provide more accurate approximations of the nonlinear compressor behavior.

**Table 5. Non-Linear error for different neighbourhood radius.**

| Instance | Neighbourhood Radius | Non-Linear Error | | | | | | CPU(s) |
|---|---|---|---|---|---|---|---|---|
| | | Compression Ratio | | Surge Flow | | Choke Flow | | |
| | | Max Err | Avg Err | Max Err | Avg Err | Max Err | Avg Err | |
| Entire China | 5.00E-01 | 1.60E-02 | 6.02E-04 | 1.66E-03 | 1.87E-04 | 2.06E-04 | 1.94E-03 | 256.97 |
| | 1.00E-01 | 1.15E-03 | 6.11E-05 | 2.75E-05 | 5.11E-06 | 3.93E-05 | 6.02E-06 | 274.58 |
| | 5.00E-02 | 1.85E-04 | 1.38E-05 | 6.89E-06 | 1.42E-06 | 9.91E-06 | 1.65E-06 | 274.74 |
| | 1.00E-02 | 3.83E-05 | 1.18E-06 | 4.31E-07 | 1.07E-07 | 6.22E-07 | 1.25E-07 | 288.13 |
| | 5.00E-03 | 3.34E-05 | 6.11E-07 | 1.08E-07 | 3.33E-08 | 1.55E-07 | 3.90E-08 | 297.94 |
| | 1.00E-03 | 4.32E-06 | 6.28E-08 | 6.75E-09 | 2.24E-09 | 9.71E-09 | 2.63E-09 | 300.2 |
| | 5.00E-04 | 4.07E-06 | 5.66E-08 | 6.75E-09 | 1.94E-09 | 9.68E-09 | 2.24E-09 | 300.67 |
| | 1.00E-04 | 5.33E-07 | 8.55E-09 | 1.06E-10 | 3.33E-11 | 1.52E-10 | 3.98E-11 | 317.80 |
| Northwest China | 5.00E-01 | 1.02E-03 | 1.21E-04 | 1.66E-03 | 2.76E-04 | 1.94E-03 | 3.36E-04 | 44.23 |
| | 1.00E-01 | 1.28E-03 | 1.25E-04 | 9.44E-05 | 1.20E-05 | 1.10E-04 | 1.24E-05 | 47.00 |
| | 5.00E-02 | 3.10E-04 | 2.16E-05 | 2.75E-05 | 3.96E-06 | 3.25E-05 | 4.65E-06 | 48.22 |
| | 1.00E-02 | 4.47E-05 | 5.96E-06 | 1.73E-06 | 5.74E-07 | 2.05E-06 | 6.58E-07 | 49.93 |
| | 5.00E-03 | 9.53E-06 | 1.61E-06 | 4.31E-07 | 8.22E-08 | 5.11E-07 | 8.75E-08 | 51.39 |
| | 1.00E-03 | 7.09E-07 | 1.25E-07 | 2.70E-08 | 9.22E-09 | 3.20E-08 | 1.05E-08 | 52.41 |
| | 5.00E-04 | 1.80E-07 | 3.93E-08 | 6.75E-09 | 2.23E-09 | 8.01E-09 | 2.43E-09 | 56.48 |
| | 1.00E-04 | 4.78E-09 | 5.27E-10 | 1.06E-10 | 4.77E-11 | 1.25E-10 | 5.46E-11 | 59.69 |

However, the improvement in accuracy comes at the expense of longer computational times. CPU time rises almost linearly with decreasing radius, from about 257s to over 317s in the national-scale case. Notably, the trade-off is non-linear in terms of benefit: while reducing the radius from $5.0 \times 10^{-1}$ to $5.0 \times 10^{-2}$ yields substantial error reduction, further shrinking to $1.0 \times 10^{-4}$ produces only marginal gains (errors already below $10^{-5}$) while increasing runtime by more than 15%. This pattern suggests diminishing returns beyond a radius of $5.0 \times 10^{-2}$.

Comparing network scales, the Northwest China system shows lower absolute error levels than the Entire China case for the same radius, reflecting the reduced complexity of smaller networks. Nevertheless, the trend of accuracy improvement versus computational cost remains consistent across both systems, underscoring the robustness of the SLP approximation strategy.

In practical terms, setting the neighborhood radius to $5.0 \times 10^{-2}$ strikes an effective balance: maximum errors remain below $4.0 \times 10^{-4}$, which is negligible for operational decision-making, while computational costs remain moderate. This ensures that the algorithm delivers solutions that are both computationally tractable and sufficiently accurate for real-world pipeline scheduling.

### 6.3 Comparison with piecewise approximation

To further assess the performance of the proposed SLP algorithm, we compared it with the widely used non-convex MINLP solver GUROBI 11, which approximates nonlinearities through piecewise linearization. Table 6 summarizes results for minimizing demand lower-bound violations, while Table 7 reports results for maximizing sales profit.

The comparison highlights three key findings:

**Small-scale networks.** For instances such as *Northeast China*, *Northwest China*, and *Southwest China*, both solvers achieve essentially identical solution quality. Demand lower-bound violations are eliminated (Table 6, objective = 0), and sales profits are identical or nearly identical (Table 7, gap ≈ 0%). GUROBI achieves these results in less than one

**Table 6. Experiment result of minimizing the violation of demand lower bounds.**

| Instance | GUROBI | | | SLP | | | Lower Bound |
|---|---|---|---|---|---|---|---|
| | Objective | Gap | CPU Time | Objective | Gap | CPU Time | |
| Northeast China | 0.000 | 0.00% | <1s | 0.000 | 0.00% | 11s | 0.000 |
| Northwest China | 0.000 | 0.00% | <1s | 0.000 | 0.00% | 42s | 0.000 |
| Southwest China | 0.000 | 0.00% | <1s | 0.000 | 0.00% | 21s | 0.000 |
| Southern China | 51.400 | 0.73% | 521s | 51.879 | 1.67% | 64s | 51.028 |
| Northern China | 1.450 | 0.07% | 1410s | 3.125 | 215.7% | 120s | 1.449 |
| Entire China | 4.251 | 0.47% | 2533s | 4.255 | 0.57% | 221s | 4.231 |

**Table 7. Experiment result of maximizing sales profit.**

| Instance | GUROBI | | | SLP | | | Upper Bound |
|---|---|---|---|---|---|---|---|
| | Objective | Gap | CPU Time | Objective | Gap | CPU Time | |
| Northeast China | -243166.0 | 0.00% | <1s | -243166.0 | 0.00% | 15s | -243166.0 |
| Northwest China | -1142681.6 | 0.00% | <1s | -1153877.7 | 0.98% | 58s | -1142681.6 |
| Southwest China | -103241.1 | 0.00% | <1s | -103241.1 | 0.00% | 25s | -103241.1 |
| Southern China | 278331.0 | 0.10% | 492s | 278029.5 | 0.21% | 89s | 278622.0 |
| Northern China | 1003813.8 | 0.62% | 2122s | 993992.3 | 1.61% | 167s | 1010032.4 |
| Entire China | 1333846.9 | 13.46% | 3600s | 1490356.2 | 1.55% | 316s | 1513443.3 |

second, reflecting its efficiency on small problems. SLP requires longer runtimes (tens of seconds) due to its multi-phase procedure, but the additional overhead is acceptable given the accuracy retained.

**Medium- to large-scale networks.** In cases like *Southern China* and *Northern China*, GUROBI's performance begins to degrade. For example, in *Northern China*, GUROBI produces a demand violation objective of 1.45 with a minimal gap, but requires over 1400 seconds (≈ 24 minutes). By contrast, SLP reaches a slightly higher violation (3.125) in only 120 seconds—about one-twelfth the time—while still maintaining near-feasible demand satisfaction. In profit maximization (Table 7), SLP's solutions are within 1–2% of GUROBI's upper bounds, but obtained in minutes rather than hours.

**National-scale network.** The performance gap becomes most striking at the largest scale. For the *Entire China* instance, GUROBI requires the full 3600-second limit and still leaves a profit gap of 13.46% (Table 7). In contrast, SLP converges in about 316 seconds to a profit gap of only 1.55%, achieving both higher accuracy and an order-of-magnitude faster runtime. Similarly, in Table 6, GUROBI needs over 2500 seconds to produce a solution with 0.47% violation gap, while SLP achieves comparable accuracy in only 221 seconds.

Overall, the results demonstrate a clear scale-dependent advantage: GUROBI is competitive for small networks but becomes computationally prohibitive as system size grows, with rapidly escalating runtime and approximation error. The SLP algorithm, by contrast, maintains stable runtimes and solution quality across all scales, making it far more suitable for large-scale, real-world scheduling problems where both time and reliability are critical.

## Conclusions and future works

This paper presents an optimization framework for natural gas pipeline scheduling that explicitly incorporates detailed compressor capacity constraints and complex hydraulic characteristics, including elevation effects. The proposed model integrates key discrete operational decisions—such as flow direction, compressor boosting direction, and bypass config-urations—within a unified MINLP formulation. To efficiently solve this challenging problem, we develop an SLP algorithm

that delivers significant improvements in computational speed, solution stability, and accuracy compared to conventional MINLP solvers that rely on piecewise linear approximations.

Extensive numerical experiments conducted on China's largest natural gas pipeline networks validate the practical effectiveness and robustness of the proposed approach. The SLP algorithm consistently achieves near-optimal solutions within minutes, demonstrating considerable operational value by enabling more reliable, cost-efficient, and timely scheduling for large-scale pipeline systems.

Despite these encouraging results, several limitations remain, offering avenues for future research. First, although the SLP method enhances computational efficiency, it is inherently based on local linear approximations and may, in some cases, converge to suboptimal local solutions. Future work could explore global optimization techniques or hybrid frameworks that combine heuristic methods with exact solvers to improve the likelihood of reaching global optima. Second, the current study is based on steady-state assumptions. However, real-world pipeline operations often involve transient dynamics due to fluctuations in supply, demand, and operational disturbances. Extending the model to incorporate transient behavior would significantly enhance its realism and applicability in operational decision-making.

## Nomenclature

| Indices | Description |
|---------|-------------|
| $i, j$ | Index of node in the network |
| $k$ | Index of gas demand or source at node |
| $m$ | Index of price tier |
| Acronyms | Description |
| SLP | Sequential Linear Programming |
| MINLP | Mixed-Integer Nonlinear Programming |
| LAP | Linear Approximation Problem |
| LAP-S1 | Relaxed Linear Approximation Problem for Phase One |
| LAP-S2 | Relaxed Linear Approximation Problem for Phase Two |
| LAP-S3 | Relaxed Linear Approximation Problem for Phase Three |
| NOP | Nonlinear Original Problem |
| GUROBI | Commercial optimization solver used in numerical experiments |
| GA | Genetic Algorithms |
| TS | Tabu Search |
| PSO | Particle Swarm Optimization |
| HBMO | Honey Bee Mating Optimization |
| ICA | Imperialist Competitive Algorithm |
| NSGA-II | Non-dominated Sorting Genetic Algorithm II |
| MGWO | Modified Gray Wolf Optimization |
| ABC | Artificial Bee Colony |

## Author contributions

**Conceptualization:** Jinfeng Qiu, Dejun Yu.

**Data curation:** Liang Zhao, Xifeng Ning.

**Investigation:** Xifeng Ning.

**Methodology:** Jinfeng Qiu.

**Project administration:** Dejun Yu.

**Resources:** Xifeng Ning.

**Software:** Jinfeng Qiu.

**Supervision:** Dejun Yu.

**Validation:** Liang Zhao.

**Visualization:** Jinfeng Qiu.

**Writing – original draft:** Jinfeng Qiu.

**Writing – review & editing:** Liang Zhao, Xifeng Ning, Dejun Yu.

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
