## [Decision Letter · Decision Letter 0]

18 Mar 2025

Dear Dr. Qiu,

Thank you for submitting your manuscript to PLOS ONE. After careful consideration, we feel that it has merit but does not fully meet PLOS ONE’s publication criteria as it currently stands. Therefore, we invite you to submit a revised version of the manuscript that addresses the points raised during the review process.

We look forward to receiving your revised manuscript.

Kind regards,

Kind regards, Soheil Mohtaram

Academic Editor

PLOS ONE

Journal Requirements:

https://journals.plos.org/plosone/s/file?id=wjVg/PLOSOne_formatting_sample_main_body.pdf   and  and

Reviewers' comments:

Reviewer's Responses to Questions

**Comments to the Author**

1. Is the manuscript technically sound, and do the data support the conclusions?

Reviewer #1: Partly

Reviewer #2: Partly

2. Has the statistical analysis been performed appropriately and rigorously?

Reviewer #1: N/A

Reviewer #2: N/A

3. Have the authors made all data underlying the findings in their manuscript fully available?

Reviewer #1: No

Reviewer #2: No

4. Is the manuscript presented in an intelligible fashion and written in standard English?

Reviewer #1: No

Reviewer #2: No

Reviewer #1: 1. The paper lacks in major contribution and motivation of research.

2. The background and significance of this study should be highlighted in the abstract.

3. Check the English presentation of this paper to remove the typo mistakes. Some grammatical issues need to be addressed in the whole text. Please reform the long paragraphs. Please polish the writing and English of the manuscript carefully. The writing of the paper needs a lot of improvement in terms of grammar, spelling, and presentation. The paper needs careful English polishing since there are many typos and poorly written sentences. I found several errors.

4. In the "Introduction" section, a more detailed analysis of the existing literature on the subject is needed, and an in-depth analysis of the possible application fields.

5. The mathematics used throughout the article is still not very strict. Please try to update and illustrate some elements in the mathematical model that are not defined very strictly.

6. Please categorize your and previous research in the Table in the section

Literature Review to show the better research gap.

7. The overall structure of the article should be improved.

8. The result part is week, results and discussion should be better explained.

9. References must be updated.

10. Check all of your Figures and Tables have a good explanation of your text.

11. Many paragraphs without citations.

12. What are the contributions and novelty of work mentioned?

13. The authors' conclusions need to be improved, a comparison of the results obtained with those already existing in the literature would be appropriate. I suggest also describing what can still be improved in this work, which can still be improved based on the results obtained, according to the authors' view. It is suggested to offer some limitations existed in this study and an outlook for future study in the last section.

Reviewer #2: General Comment:

The manuscript presents a promising study; however, several areas need improvement to enhance its clarity, readability, and scientific contribution. The authors should address issues related to language quality, novelty presentation, and proper referencing. Additionally, improving the figures, expanding the literature review, and providing more detail in the conclusions section would strengthen the manuscript significantly. Below are more specific comments to guide the authors in revising the paper.

1. Language Quality:

o The manuscript would benefit from a thorough review of its English language usage to improve clarity and readability. It is recommended that the authors address grammatical and syntactic errors throughout the text to enhance the quality of the writing.

2. Novelty and Original Contribution:

o The novelty of the research is not clearly highlighted. The authors should explicitly articulate the unique contributions of their work to distinguish it from existing studies.

3. Nomenclature Section:

o To improve the manuscript's readability, the authors are encouraged to include a nomenclature section at the beginning. This should contain all variables, acronyms, indices, and constants used in the manuscript, allowing readers to refer easily to these definitions.

4. Introduction and Literature Review:

o The Introduction would benefit from a more thorough review of relevant literature. A comprehensive manuscript in this domain typically references a minimum of 40 studies. Expanding the literature review will also provide a stronger foundation for the study's relevance and innovation.

5. Recommended References:

o The following references are highly relevant to the subject and should be incorporated to enhance the background and context of the study:

Detection of internal fault in differential transformer protection based on fuzzy method. International Journal of Physical Sciences, 6(26), pp.6150-6158, 2011.

Distributed generation management in smart grid with the participation of electric vehicles with respect to the vehicle owners’ opinion by using the imperialist competitive algorithm. Sustainability, 14(8), p.4770, 2022.

Optimum operation management of microgrids with cost and environment pollution reduction approach considering uncertainty using multi‐objective NSGAII algorithm. IET Renewable Power Generation, 2022.

Economic dispatch optimization considering operation cost and environmental constraints using the HBMO method. Energy Reports, 10, pp.1718-1725, 2023.

Unit commitment for power generation systems based on prices in smart grid environment considering uncertainty. Sustainability, 13(18), p.10219, 2021.

Locating and sizing of capacitor banks and multiple DGs in distribution system to improve reliability indexes and reduce loss using ABC algorithm. Bulletin of Electrical Engineering and Informatics, 10(2), pp.559-568, 2021.

Comparison of SVC and STATCOM in static voltage stability margin enhancement. International Journal of Electrical and Computer Engineering, 3(2), pp.297-302, 2009.

Optimal management of energy storage systems for peak shaving in a smart grid. Computers, Materials and Continua, 75(2), pp.3317-3337, 2023.

Optimal design of solar–wind hybrid system-connected to the network with cost-saving approach and improved network reliability index. SN Applied Sciences, 1(12), p.1742, 2019.

6. Figures Quality:

o Figures in the manuscript should be provided in higher resolution to improve clarity. Each figure should also be thoroughly described within the text to ensure that it contributes effectively to the manuscript's presentation.

7. Future Work and Limitations:

o A brief discussion of potential limitations and directions for future research should be included in the Conclusions and Recommendations section. This will provide a more comprehensive view of the study’s scope and areas for future exploration.

**Do you want your identity to be public for this peer review?** For information about this choice, including consent withdrawal, please see our For information about this choice, including consent withdrawal, please see our Privacy Policy .

Reviewer #1: No

Reviewer #2: No

While revising your submission, please upload your figure files to the Preflight Analysis and Conversion Engine (PACE) digital diagnostic tool, https://pacev2.apexcovantage.com/ . PACE helps ensure that figures meet PLOS requirements. To use PACE, you must first register as a user. Registration is free. Then, login and navigate to the UPLOAD tab, where you will find detailed instructions on how to use the tool. If you encounter any issues or have any questions when using PACE, please email PLOS at . PACE helps ensure that figures meet PLOS requirements. To use PACE, you must first register as a user. Registration is free. Then, login and navigate to the UPLOAD tab, where you will find detailed instructions on how to use the tool. If you encounter any issues or have any questions when using PACE, please email PLOS at figures@plos.org . Please note that Supporting Information files do not need this step.. Please note that Supporting Information files do not need this step.

---

## [Author Response · Author response to Decision Letter 1]

19 May 2025

First and foremost, we want to thank the editor and the reviewer for your careful review and constructive comments of our manuscript. We have read the comments carefully and revised our paper accordingly. Our point-to-point responses to the comments are provided below. For the ease of reading, all of our responses are in blue font color, and are started with [Authors]. Any inserted, replaced or modified texts to the manuscript are in italics in this document. In addition, we use the red-colored font to highlight some important changes/modifications in the revised manuscript.

Reviewer #1:

1. The paper lacks in major contribution and motivation of research.

[Authors] Thanks for the comment. In the revised manuscript, we have thoroughly reorganized and rewritten the Introduction section to better highlight the motivation behind our work and to clearly position it within the context of existing research. Furthermore, we have added a new Section 2.6 titled "Our Contributions", where we explicitly summarize and clarify the major contributions of this study.

2. The background and significance of this study should be highlighted in the abstract.

[Authors] Thanks for the comment. We have carefully reorganized and rewritten the Abstract section to better emphasize the background and significance of our research. The revised abstract now provides a clearer context of the problem being addressed and highlights the importance of our study in the field.

3. Check the English presentation of this paper to remove the typo mistakes. Some grammatical issues need to be addressed in the whole text. Please reform the long paragraphs. Please polish the writing and English of the manuscript carefully. The writing of the paper needs a lot of improvement in terms of grammar, spelling, and presentation. The paper needs careful English polishing since there are many typos and poorly written sentences. I found several errors.

[Authors] Thank you for pointing this out. We have carefully reviewed the entire manuscript and made comprehensive revisions to improve the English presentation. This includes correcting grammatical errors, eliminating typos, restructuring overly long paragraphs, and enhancing the overall clarity and readability of the text. We have polished the writing throughout the manuscript to ensure it meets the standards of academic English.

4. In the "Introduction" section, a more detailed analysis of the existing literature on the subject is needed, and an in-depth analysis of the possible application fields.

[Authors] Thanks for the comment. We have expanded the literature review and conducted a more detailed analysis of the existing studies in Section 2: Literature Review. This revision not only provides a clearer overview of the current research landscape but also discusses the potential application fields of our study in greater depth.

5. The mathematics used throughout the article is still not very strict. Please try to update and illustrate some elements in the mathematical model that are not defined very strictly.

[Authors] Thanks for the comment. In the revised version of the manuscript, we have carefully reviewed the mathematical formulations and made the necessary modifications to ensure greater rigor and clarity. Specifically, we have refined the definitions of key elements and improved the consistency and accuracy of the notations used in our model.

6. Please categorize your and previous research in the Table in the section

Literature Review to show the better research gap.

[Authors] Thanks for the comment. Thank you for your suggestion. While we did not add a table, we have revised Section 2: Literature Review to include a more detailed comparative analysis between previous studies and our own research. This enhanced discussion more clearly outlines the differences in methodology, focus, and contributions, thereby better highlighting the research gap that our work addresses.

7. The overall structure of the article should be improved.

[Authors] Thank you for your suggestion. In the revised manuscript, we have reorganized the overall structure of the paper to improve its clarity and logical flow. The updated structure now consists of seven sections: Introduction, Literature Review, Problem Description, The Mathematical Programming Model, Solution Algorithm, Numerical Experiment, Conclusions and Future Works.

8. The result part is week, results and discussion should be better explained.

[Authors] Thanks for the comment. In the revised manuscript, we have enriched the Numerical Experiment section by providing more detailed explanations of the experimental results and further elaborating on their implications. We also enhanced the discussion to better interpret the findings, compare them with existing methods, and highlight the strengths and limitations of our approach.

9. References must be updated.

[Authors] Thanks for the comment. We have updated the reference list by adding and incorporating more recent and relevant literature to reflect the current state of research in this field.

10. Check all of your Figures and Tables have a good explanation of your text.

[Authors] Thanks for the comment. We have carefully checked and revised all the figures and tables in the manuscript to ensure that each one is clearly explained and well-integrated into the main text. We also revised the corresponding captions and descriptions to improve clarity and consistency, making it easier for readers to understand their relevance and role in supporting our research findings.

11. Many paragraphs without citations.

[Authors] Thank you for pointing this out. We have thoroughly reviewed the manuscript and revised the relevant sections to ensure that appropriate citations are provided wherever necessary.

12. What are the contributions and novelty of work mentioned?

[Authors] Thanks for the comment. To clearly present the contributions and novelty of our work, we have added a new Section 2.7 titled "Our Contributions" in the revised manuscript. In this section, we explicitly summarize the key innovations and contributions of our research.

13. The authors' conclusions need to be improved, a comparison of the results obtained with those already existing in the literature would be appropriate. I suggest also describing what can still be improved in this work, which can still be improved based on the results obtained, according to the authors' view. It is suggested to offer some limitations existed in this study and an outlook for future study in the last section.

[Authors] Thanks for the comment. In the revised manuscript, we have enhanced the Conclusions and Future Works section by providing a more comprehensive summary of our key findings and their implications. Furthermore, we have added a critical discussion of the limitations of the current study and outlined several potential directions for future research based on the observed results.

Reviewer #2: General Comment:

The manuscript presents a promising study; however, several areas need improvement to enhance its clarity, readability, and scientific contribution. The authors should address issues related to language quality, novelty presentation, and proper referencing. Additionally, improving the figures, expanding the literature review, and providing more detail in the conclusions section would strengthen the manuscript significantly. Below are more specific comments to guide the authors in revising the paper.

[Authors] We sincerely thank the reviewer for the positive recognition of the potential value of our study, as well as for the detailed and constructive feedback. We will address your suggestions as follows:

1. Language Quality:

o The manuscript would benefit from a thorough review of its English language usage to improve clarity and readability. It is recommended that the authors address grammatical and syntactic errors throughout the text to enhance the quality of the writing.

[Authors] Thanks for the comment. We have carefully reviewed and revised the entire manuscript to correct grammatical, syntactic, and typographical errors. These revisions were made to improve the overall clarity, readability, and quality of the writing.

2. Novelty and Original Contribution:

o The novelty of the research is not clearly highlighted. The authors should explicitly articulate the unique contributions of their work to distinguish it from existing studies.

[Authors] Thanks for the comment. We have added a new section (Section 2.6: Our Contributions) in the revised manuscript. This section clearly summarizes the key innovations and original contributions of our research in comparison to existing methods.

3. Nomenclature Section:

o To improve the manuscript's readability, the authors are encouraged to include a nomenclature section at the beginning. This should contain all variables, acronyms, indices, and constants used in the manuscript, allowing readers to refer easily to these definitions.

[Authors] Thank you for the suggestion. In the revised manuscript, we have added a list of acronyms and abbreviations at the beginning to enhance readability. Additionally, we have provided detailed definitions and classifications of all key variables, parameters, and indices in Tables 1 to 3.

4. Introduction and Literature Review:

o The Introduction would benefit from a more thorough review of relevant literature. A comprehensive manuscript in this domain typically references a minimum of 40 studies. Expanding the literature review will also provide a stronger foundation for the study's relevance and innovation.

[Authors] Thanks for the comment. In the revised manuscript, we have expanded Section 2: Literature Review to include a broader and more comprehensive survey of relevant studies in this field. This enhancement helps to better position our work within the existing body of literature and to strengthen the foundation for the research’s relevance and innovation. The number of references has also been increased accordingly to reflect the depth of the revised review.

5. Recommended References:

o The following references are highly relevant to the subject and should be incorporated to enhance the background and context of the study:

Detection of internal fault in differential transformer protection based on fuzzy method. International Journal of Physical Sciences, 6(26), pp.6150-6158, 2011.

Distributed generation management in smart grid with the participation of electric vehicles with respect to the vehicle owners’ opinion by using the imperialist competitive algorithm. Sustainability, 14(8), p.4770, 2022.

Optimum operation management of microgrids with cost and environment pollution reduction approach considering uncertainty using multi‐objective NSGAII algorithm. IET Renewable Power Generation, 2022.

Economic dispatch optimization considering operation cost and environmental constraints using the HBMO method. Energy Reports, 10, pp.1718-1725, 2023.

Unit commitment for power generation systems based on prices in smart grid environment considering uncertainty. Sustainability, 13(18), p.10219, 2021.

Locating and sizing of capacitor banks and multiple DGs in distribution system to improve reliability indexes and reduce loss using ABC algorithm. Bulletin of Electrical Engineering and Informatics, 10(2), pp.559-568, 2021.

Comparison of SVC and STATCOM in static voltage stability margin enhancement. International Journal of Electrical and Computer Engineering, 3(2), pp.297-302, 2009.

Optimal management of energy storage systems for peak shaving in a smart grid. Computers, Materials and Continua, 75(2), pp.3317-3337, 2023.

Optimal design of solar–wind hybrid system-connected to the network with cost-saving approach and improved network reliability index. SN Applied Sciences, 1(12), p.1742, 2019.

[Authors] Thanks for the comment. All of the suggested references have been carefully reviewed and incorporated into the Literature Review section of the revised manuscript, as they provide valuable background and context for our study.

6. Figures Quality:

o Figures in the manuscript should be provided in higher resolution to improve clarity. Each figure should also be thoroughly described within the text to ensure that it contributes effectively to the manuscript's presentation.

[Authors] Thanks for the comment. In the revised manuscript, we have replaced the figures with higher-resolution versions to improve clarity and ensure that they meet the quality standards. Additionally, we have provided more detailed descriptions of each figure within the text to ensure that they effectively contribute to the manuscript's presentation and enhance the understanding of the results.

7. Future Work and Limitations:

o A brief discussion of potential limitations and directions for future research should be included in the Conclusions and Recommendations section. This will provide a more comprehensive view of the study’s scope and areas for future exploration.

[Authors] Thanks for the comment. In response, we have expanded the Conclusions and Future Works section to include a brief discussion of the limitations of our current study. We have also outlined several potential directions for future research based on the findings of this work.

---

## [Decision Letter · Decision Letter 1]

22 Jul 2025

Dear Dr. Yu,

Thank you for submitting your manuscript to PLOS ONE. After careful consideration, we feel that it has merit but does not fully meet PLOS ONE’s publication criteria as it currently stands. Therefore, we invite you to submit a revised version of the manuscript that addresses the points raised during the review process.

We look forward to receiving your revised manuscript.

Kind regards,

Soheil Mohtaram

Academic Editor

PLOS ONE

Journal Requirements:

Reviewers' comments:

Reviewer's Responses to Questions

**Comments to the Author**

Reviewer #1: All comments have been addressed

Reviewer #3: All comments have been addressed

Reviewer #4: All comments have been addressed

2. Is the manuscript technically sound, and do the data support the conclusions?

Reviewer #1: Partly

Reviewer #3: Yes

Reviewer #4: Yes

3. Has the statistical analysis been performed appropriately and rigorously?

Reviewer #1: No

Reviewer #3: Yes

Reviewer #4: Yes

4. Have the authors made all data underlying the findings in their manuscript fully available?

Reviewer #1: No

Reviewer #3: Yes

Reviewer #4: Yes

5. Is the manuscript presented in an intelligible fashion and written in standard English?

Reviewer #1: No

Reviewer #3: Yes

Reviewer #4: Yes

Reviewer #1: 1. The writing of the paper still needs careful English polishing since there are many typos and poorly written sentences. I found several errors.

2. The overall structure of the article should be improved.

3. The result part still week, results and discussion must be better explained.

4. Add the suitable references from the following

https://doi.org/10.1007/s13369-023-07975-7
https://doi.org/10.1109/MEPCON58725.2023.10462453
https://doi.org/10.1038/s41598-024-78384-5
https://doi.org/10.1109/ACCESS.2024.3356556
https://doi.org/10.1109/ACCESS.2024.3437191 10.21608/SVUSRC.2023.240888.1153, https://doi.org/10.1109/EIConRus.2018.8317170
https://doi.org/10.3390/en12050961
https://doi.org/10.1109/MEPCON.2017.8301313
https://doi.org/10.1007/s00521-024-09433-3
https://doi.org/10.20508/ijrer.v14i2.14346.g8898
https://doi.org/10.1371/journal.pone.0317619
https://dx.doi.org/10.21608/jaet.2021.82231
https://doi.org/10.1109/MEPCON47431.2019.9008171

Reviewer #3: The responses demonstrate a strong understanding of the reviewers' concerns and a clear plan of action taken in the revised manuscript. The authors frequently indicate specific sections where changes have been made.

Reviewer #4: The paper tackles a problem of immense practical and economic importance. The proposed solution method is pragmatic, and the results are compelling. The work is well-structured and demonstrates a strong understanding of both the physical system and the underlying optimization challenges. While the core technical contribution is sound, the manuscript requires significant revisions, primarily related to manuscript preparation and clarity, before it can be considered for publication.

1. The manuscript is generally well-written, but it suffers from several critical formatting errors and areas where clarity could be improved.

I. The literature review is replete with broken citations, appearing as "Error! Reference source not found." This must be corrected throughout the entire manuscript.

II. There is a significant contradiction regarding data availability. The submission form states, "All data of gas networks are available from the github

III. https://github.com/masonYau/gasNetworkData/tree/main". However, the manuscript text states, "The experimental data in the paper can be obtained by contacting Mr. Yu Dejun". This inconsistency must be resolved. For transparency and reproducibility, the GitHub repository is the far superior option and should be the sole method cited.

IV. Some sentences could be more concise. For example, "The compression ratio a compressor can provide is intricately linked to its operating conditions... and involves complex higher-order nonlinear physical relationships” could be simplified to "A compressor's compression ratio is a complex, nonlinear function of its operating conditions, including inlet flow rate and rotational speed."

**Do you want your identity to be public for this peer review?** For information about this choice, including consent withdrawal, please see our For information about this choice, including consent withdrawal, please see our Privacy Policy .

Reviewer #1: No

Reviewer #3: No

Reviewer #4: No

While revising your submission, please upload your figure files to the Preflight Analysis and Conversion Engine (PACE) digital diagnostic tool, https://pacev2.apexcovantage.com/ . PACE helps ensure that figures meet PLOS requirements. To use PACE, you must first register as a user. Registration is free. Then, login and navigate to the UPLOAD tab, where you will find detailed instructions on how to use the tool. If you encounter any issues or have any questions when using PACE, please email PLOS at . PACE helps ensure that figures meet PLOS requirements. To use PACE, you must first register as a user. Registration is free. Then, login and navigate to the UPLOAD tab, where you will find detailed instructions on how to use the tool. If you encounter any issues or have any questions when using PACE, please email PLOS at figures@plos.org . Please note that Supporting Information files do not need this step.. Please note that Supporting Information files do not need this step.

---

## [Author Response · Author response to Decision Letter 2]

3 Sep 2025

Response to Reviewers

Manuscript ID: PONE-D-25-04544R1

Title: Gas network value chain optimization considering compressor capacity constraints

Dear Academic Editor and Reviewers,

We sincerely thank the Academic Editor and all reviewers for their constructive comments and valuable suggestions, which have significantly helped us improve the quality and clarity of our manuscript. We have carefully revised the manuscript accordingly. Below, we provide a detailed, point-by-point response to each reviewer’s comments. Changes have been incorporated in the revised manuscript and highlighted accordingly.

Reviewer #1

Comment 1: The writing of the paper still needs careful English polishing since there are many typos and poorly written sentences. I found several errors.

Response: We appreciate this important comment. The manuscript has undergone thorough English editing and proofreading to improve readability, grammar, and style. Long and complex sentences have been simplified, and typographical errors have been corrected.

Comment 2: The overall structure of the article should be improved.

Response: We have revised the structure of the manuscript to improve clarity and logical flow. In particular, the Introduction has been streamlined to emphasize research motivation and contributions, and the Literature Review has been reorganized into thematic subsections.

Comment 3: The result part still week, results and discussion must be better explained.

Response: We reorganized explanatory paragraphs after each major results table, clarifying the accuracy of the SLP approach, its scalability compared to GUROBI, and the impact of compressor capacity constraints.

Comment 4: Add the suitable references from the following list.

Response: We carefully reviewed the suggested references and incorporated several recent and relevant works into Section 2.6 Emerging Trends in Energy Network Optimization. For example, we added discussions of studies on microgrid energy management (Abdelsattar et al., 2023; Abdelsattar et al., 2024a), machine learning for renewable energy forecasting (Abdelsattar et al., 2024b; Abdelsattar et al., 2025), and metaheuristic algorithms for hybrid energy systems (Abdelsattar et al., 2024c). These additions strengthen the context of our work and highlight its contribution by addressing compressor capacity constraints in natural gas networks.

Reviewer #3

Comment: The responses demonstrate a strong understanding of the reviewers' concerns and a clear plan of action taken in the revised manuscript. The authors frequently indicate specific sections where changes have been made.

Response: We sincerely thank the reviewer for their positive assessment. We have maintained this careful revision strategy in the current version and ensured that all changes are clearly indicated in the revised manuscript.

Reviewer #4

Comment 1: The manuscript suffers from several critical formatting errors and areas where clarity could be improved.

Response: We have corrected all formatting issues. In particular, broken citations previously appearing as “Error! Reference source not found.” have been fixed, and consistency in headings, figures, and tables has been ensured.

Comment 2: There is a significant contradiction regarding data availability. The submission form states GitHub availability, but the text says “contact the author.” This inconsistency must be resolved.

Response: We thank the reviewer for pointing out this contradiction. We have revised the Data Availability statement to unify the description. All data are now clearly stated as being openly available on GitHub:

https://github.com/masonYau/gasNetworkData/tree/main

Comment 3: Some sentences could be more concise.

Response: We have simplified several overly complex sentences throughout the manuscript. For example, the sentence about compressor compression ratio was revised to:

“A compressor’s compression ratio is a complex, nonlinear function of its operating conditions, including inlet flow rate and rotational speed.”

This makes the text clearer and more concise.

Final Remarks

We believe that the revised manuscript has significantly improved in terms of language, structure, results interpretation, and consistency. We thank the reviewers again for their helpful feedback and look forward to your favorable consideration.

---

## [Editor Report · Decision Letter 2]

5 Sep 2025

Gas network value chain optimization considering compressor capacity constraints

PONE-D-25-04544R2

Dear Dr. Dejun Yu,

We’re pleased to inform you that your manuscript has been judged scientifically suitable for publication and will be formally accepted for publication once it meets all outstanding technical requirements.

Kind regards,

Soheil Mohtaram

Academic Editor

PLOS ONE

---

## [Editor Report · Acceptance letter]

PONE-D-25-04544R2

PLOS One

Dear Dr. Yu,

I'm pleased to inform you that your manuscript has been deemed suitable for publication in PLOS One. Congratulations! Your manuscript is now being handed over to our production team.

Kind regards,

on behalf of

Dr. Soheil Mohtaram

Academic Editor

PLOS One